# Graphene-Supported Cu*_n_* (*n* = 5, 6) Clusters for CO_2_ Reduction Catalysis

**DOI:** 10.3390/nano15060445

**Published:** 2025-03-15

**Authors:** Yanling Guo, Lisu Zhang, Yanbo Zou, Xingguo Wang, Qian Ning

**Affiliations:** 1School of Physics and Electronic Engineering, Xinjiang Normal University, Urumqi 830054, China; guoyl20000508@163.com (Y.G.);; 2Xinjiang Key Laboratory for Luminescence Minerals and Optical Functional Materials, Urumqi 830054, China; 3College of Electronics and Information Engineering, Sichuan University, Chengdu 610065, China

**Keywords:** electrocatalytic CO_2_ reduction reaction, Cu cluster, composite catalyst, first-principles calculations

## Abstract

In recent years, driven by the swift progress in nanotechnology and catalytic science, researchers in the field of physical chemistry have been vigorously exploring novel catalysts designed to enhance the efficiency and selectivity of a broad spectrum of chemical reactions. Against this backdrop, Cu clusters supported on defective graphene (Cu*_n_*@GR, where *n* = 5, 6) function as two-dimensional nanocatalysts, demonstrating exceptional catalytic activity in the electrochemical reduction of carbon dioxide (CO_2_RR). A comprehensive investigation into the catalytic properties of these materials has been undertaken using density functional theory (DFT) calculations. By tailoring the configuration of Cu*_n_*@GR, specific reduction products such as CH_4_ and CH_3_OH can be selectively produced. The product selectivity is quantitatively analyzed through free energy calculations. Remarkably, the Cu_5_@GR catalyst enables the electrochemical reduction of CO_2_ to CH_4_ with a significantly low overpotential of −0.31 eV. Furthermore, the overpotential of the hydrogen evolution reaction (HER) is higher than that of the conversion of CO_2_ to CH_4_; hence, the HER is unlikely to interfere and impede the efficiency of CH_4_ production. This study demonstrates that Cu_5_@GR offers low overpotential and high catalytic efficiency, providing a theoretical foundation for the design and experimental synthesis of composite nanocatalysts.

## 1. Introduction

Globally, 80% of current energy consumption continues to depend on fossil fuels. The burning of these resources emits significant amounts of carbon dioxide (CO_2_), intensifying the impacts of climate change and accelerating global warming. In the drive toward carbon neutrality, there is an urgent need to reduce atmospheric CO_2_ concentrations. This critical challenge has spurred researchers worldwide to explore and develop innovative strategies for mitigating, capturing, and valorizing CO_2_.

A promising approach involves converting CO_2_ into economically valuable chemicals, such as methanol and methane, to achieve a sustainable carbon cycle. Current technologies include photochemical reduction, electrochemical reduction, and bioreduction processes. Remarkably, the Cu_5_@GR catalyst enables the electrochemical reduction of CO_2_ to CH_4_ with a significantly low overpotential of −0.31 eV. In particular, the electrochemical CO_2_ reduction reaction (CO_2_RR) has attracted considerable interest owing to its exceptional conversion efficiency and rapid reaction kinetics.

The electrocatalytic reduction of CO_2_ presents a significant challenge. As a linear, non-polar molecule with sp orbital hybridization, CO_2_ features two oxygen atoms that form stable C=O double bonds with the central carbon atom. This structural configuration confers CO_2_ with remarkable thermodynamic stability and chemical inertness, rendering its reduction process highly challenging. Extensive theoretical and experimental investigations have shown that suitable catalysts can efficiently facilitate CO_2_ activation [1,2]. Consequently, the development of highly efficient CO_2_RR catalysts with high product selectivity and low overpotential has become a major research focus.

Cluster catalysts have emerged as a prominent research focus in catalysis, owing to their high atomic utilization efficiency, maximized metal–support interface, and enhanced activity ratio. Experimental studies have demonstrated that Au and Pd nanoparticles exhibit high CO generation efficiency [3,4]. When nitrogen-doped carbon-supported Au_2_ clusters are employed as catalysts for CO_2_RR, the primary products are CO and H_2_ [5]. The limiting potentials required for reducing CO_2_ to CH_4_ using Fe_2_@BN and Ni_2_@BN are as low as −0.47 V and −0.39 V, respectively [6]. Among various electrocatalysts, copper (Cu) is the only known electrochemical catalyst capable of converting CO_2_ into alternative energy fuels and hydrocarbons, particularly methane (CH_4_), with adequate current density and selectivity [7]. However, several challenges remain, including the issue of product mixture, the competing HER, and the high overpotential required for CO_2_RR on monometallic Cu. Therefore, the development of Cu-based catalysts with high selectivity and low overpotential has become an area of significant research interest [8].

Numerous studies have reported the exceptional performance of small-sized Cu clusters in CO_2_RR. Suitable substrates can enhance the stability of these clusters and modulate their catalytic properties. In 2022, Du et al. calculated that the overpotential of Cu_3_ clusters loaded onto defective graphene during the electrochemical reduction of CO_2_ to CH_4_ was only 0.53 eV [9]. In 2024, Pan et al. successfully prepared a carbon support with a continuous dual-pore structure by precisely controlling the atomic-scale arrangement of the Cu cluster active centers and ingeniously designing the mesoscale structure of the carbon support. They embedded N and OH-modified Cu_3_ clusters into this bicontinuous carbon mesoporous system. In CO_2_RR experiments, this catalyst exhibited excellent performance, achieving a Faraday efficiency of 74.2% for CH_4_ at a current density of 300 mA cm^−2^. The overpotential for the reaction was calculated to be 0.85 eV based on free energy calculations [10]. Despite recent advances, existing catalysts still face the challenge of high overpotentials, which not only increase energy input but also compromise catalyst stability by leading to higher current densities, thus accelerating degradation and structural changes. Reducing the overpotential can enhance energy efficiency and prolong catalyst lifespan by mitigating these adverse effects and improving overall stability. Therefore, further investigation into the rate-limiting steps and reaction pathways is crucial to developing more efficient catalysts with lower overpotentials.

Building on recent research, we investigated the catalytic efficiency of Cu_5_ (Cu_5_@GR) and Cu_6_ (Cu_6_@GR) cluster catalysts supported on defective graphene in the CO_2_RR process. Computational results demonstrate that Cu_5_@GR and Cu_6_@GR catalysts effectively promote the deep reduction of CO_2_, generating non-toxic monatomic products, such as CH_4_, under low potential conditions. Notably, Cu_n_ clusters possessing analogous active sites exhibit uniform reduction products and conversion pathways.

## 2. Calculation Method

All first-principles calculations were performed using density functional theory (DFT) [11] within the Vienna Ab Initio Simulation Package (VASP) [12,13], utilizing the Projector Augmented Wave (PAW) method [14] to account for core and valence electron interactions. The Perdew–Burke–Ernzerhof (PBE) functional within the generalized gradient approximation (GGA) [15] was employed for geometric optimization, incorporating transformation correlation [16]. The plane-wave cut-off energy was set to 400 eV to ensure convergence. For all graphene-supported structures, a 5 × 5 × 1 primitive graphene supercell was constructed, followed by the incorporation of Cu clusters. The lattice dimensions of 21.31 Å in length and 12.34 Å in width ensured the adequate dispersion of Cu clusters, minimizing electronic interactions between clusters that could interfere with catalytic activity. A vacuum layer of 15 Å was introduced along the z-direction to eliminate periodic interactions. A 5 × 5 × 1 Monkhorst–Pack grid [17] was used to sample the reciprocal space, and all geometries were fully relaxed until the maximum residual force in all directions was reduced to below 0.02 eV/Å. The electronic energy was minimized to a tolerance of 10^−5^ eV, consistent with the structural parameters of Cu dimers anchored on graphene [18].

In this calculation model, the adsorption energy (*E*_ads_) of CO_2_ and intermediates can be calculated as*E*_ads_ = *E*_AB_ − *E*_A_ − *E*_B_
(1)
where *E*_AB_ represents the total energy of the product and *E*_A_ and *E*_B_ are the energies of the reactants.

The Gibbs free energy difference (Δ*G*) can be calculated asΔ*G* = Δ*E*_elec_ + Δ*E*_zpe_ − *T*ΔS (2)
where Δ*E*_elec_ is the reaction energy from the DFT total energies. In addition, we calculated zero-point energy correction Δ*E*_zpe_ and entropy energy correction *T*ΔS of adsorbates according to the quantum mechanical harmonic approximation at 298.15 K.

## 3. Results and Discussion

### 3.1. Catalyst Structure

Defect sites on the graphene surface can be occupied by Cu atoms, facilitating the high dispersion of Cu species on the substrate. Experimentally, by adjusting the reduction temperature of Cu deposited on the graphite carbon shell, different types of Cu*_n_*@GR (*n* = 5, 6) catalysts [19] can be synthesized. These defect sites on the graphene surface serve as anchoring points for the Cu atoms, as shown in Figure 1. Optimized Cu_5_ and Cu_6_ clusters were placed near these defect sites to disperse the Cu*_n_* clusters (*n* = 5, 6) across the surface, forming the Cu*_n_*@GR structures, as depicted in Figure 2. The initial structures of the Cu*_n_*@GR catalysts with different configurations were optimized, and their binding energies were calculated to identify the most stable configuration for electrocatalytic studies. Among the Cu_5_@GR configurations, the optimal structure is Cu_5_@GR-II, while for Cu_6_@GR, the optimal structure is Cu_6_@GR-II. In the Cu*_n_* clusters, Cu atoms are tightly bonded through metallic interactions, while the anchored Cu atoms interact with heteroatoms from the defective graphene support. As shown in Figure 3, the binding between the Cu*_n_* clusters and the support is exothermic and spontaneous in all cases, with the binding strength between carbon and Cu atoms sufficient to anchor the Cu*_n_* (*n* = 5, 6) clusters on the substrate. Theoretically, the configuration with the lowest binding energy is expected to exhibit the highest presence ratio and superior stability during the experimental preparation of metal clusters. Therefore, subsequent research will focus solely on the optimal configuration with the lowest binding energy.

The optimized configuration of ELFCu*_n_*@GR, as shown in Figure 4, demonstrates strong coupling between Cu-C bonds. The partial density of states (PDOS) provides a clear explanation for the structural stability of the catalyst, with the PDOS of Cu*_n_*@GR illustrated in Figure 5. Near the Fermi level, a pronounced overlap is observed between the d-orbitals of Cu atoms and the p-orbitals of C atoms. This orbital overlap indicates the formation of covalent interactions between the metal atoms and their adjacent atoms. A further analysis of the PDOS reveals that the metal clusters are stably anchored on the defect-doped graphene substrate. Notably, the PDOS peaks of the Cu clusters and the substrate exhibit a high degree of overlap near the Fermi level, providing compelling evidence for the strong interaction between the Cu*_n_* clusters and the substrate. 

### 3.2. Catalytic Activity

To explore the catalytic activity of Cu*_n_*@GR towards CO_2_, we conducted calculations to determine the charges associated with CO_2_ before and after its adsorption, with the results summarized in Table 1. Upon comparison, it is evident that post-adsorption, CO_2_ carries a higher charge compared to its pre-adsorption state. This suggests that CO_2_ is capable of acquiring additional electrons from the catalyst surface, facilitating its engagement in subsequent chemical reactions.

### 3.3. Reaction Site

In contrast to pure Cu surfaces, Cu clusters possess a higher density of under-coordinated sites at edges and corners, leading to enhanced adsorption properties and catalytic performance [20]. The catalyst’s ability to adsorb key intermediates such as *CO in CO_2_RR can effectively reflect the catalytic trend and reduction products. As shown in Figure 6, Cu atoms bonded to the substrate display higher adsorption energy for *CO than those at top interface positions, making them effective reaction centers for subsequent CO_2_RR processes. To deepen the understanding of *CO on the Cu*_n_*@GR surface, an analysis of the PDOS of the *CO species adsorbed on the catalyst was conducted, as depicted in Figure 7. The resulting spectra offer clear evidence of the electronic structural characteristics of the *CO species on Cu*_n_*@GR. Notably, significant hybridization was observed between the sp orbitals of the adsorbed *CO and the d orbitals of Cu near the Fermi level. This discovery indicates a strong interaction between *CO and Cu*_n_*@GR, facilitating the stable adsorption of *CO onto the Cu*_n_*@GR surface. Such stable adsorption provides a solid foundation for subsequent hydrogenation and reduction processes, ultimately enabling the conversion of *CO into carbon-based materials.

### 3.4. Catalytic Process

#### 3.4.1. First Step of Protonation

The initial protonation step in CO_2_ reduction can be categorized into two pathways, depending on the distinct carbon and oxygen binding sites:Path (i) *COOH: *+CO_2_ + H^+^ + e^−^ → *COOHPath (ii) *OCHO: *+CO_2_ + H^+^ + e^−^ → *OCHO

The first step of protonation involves different intermediates the subsequent reduction pathways and C_1_ products will vary. To explore Cu*_n_*@GR, the selectivity of generating the C_1_ reduction equation was calculated by separately calculating the ∆G of the catalyst adsorption of *COOH intermediate and *OCHO intermediate, as shown in Figure 8. The first step of the reaction between the two catalysts is targeting *COOH, indicating a stronger interaction between *COOH and the catalyst. Therefore, when Cu*_n_*@GR catalyzes CO_2_RR, HCOOH is not the primary product”.

#### 3.4.2. Reaction Pathway

Based on this, we evaluated the Gibbs free energies of the reaction intermediates to determine the optimal reaction pathway that requires the minimum external voltage. The entire reaction process involves eight (H^+^ + e^−^) pair transfer steps. From the free energy diagram for the electrochemical reduction of CO_2_ to CH_4_ (Figure 9), we can observe that on the two catalysts, the limiting potentials are 0.316 eV and 0.329 eV, respectively, and the rate-limiting steps occur during the conversions from *CHO to *CH_2_O and from *CH_2_O to *CH_3_O. The *CH_3_O intermediate plays a pivotal role in this process, with its hydrogenation step leading to two distinct products: one is the methanol precursor CH_3_OH, and the other is its decomposition into *O + CH_4_. It is noteworthy that under the influence of the catalyst, the conversion of *CH_3_O to *CH_3_OH exhibits endothermic characteristics, implying that it requires the absorption of energy. Conversely, the pathway for *CH_3_O decomposition into *O + CH_4_ is accompanied by a decrease in free energy, manifesting as an exothermic reaction. Given these differences in energy changes, the CO_2_RR process is more inclined to proceed in the direction of methane production; that is, *CH_3_O has a greater tendency to convert into *O + CH4 rather than CH_3_OH. These results indicate that Cu_5_@GR and Cu_6_@GR exhibit minimal energy barriers for methane formation, demonstrating that the Cu*_n_*@GR catalysts possess excellent methane production capabilities. The rate-limiting step in methane production catalyzed by Cu_5_@GR entails the hydrogenation of *CHO to *CH_2_O, necessitating a limiting potential of approximately 0.316 eV. This compares favorably to the significantly higher 0.99 eV required by the periodic Cu(111) surface, representing a reduction of 0.674 eV [21], and is also lower than the potential required by Cu_3_@GR by 0.214 eV [9]. Such findings suggest that Cu_5_@GR exhibits superior catalytic efficiency and reduced energy demands during this pivotal reaction step.

### 3.5. Analysis of Side Reactions

#### Hydrogen Evolution Reaction

Experimental observations have shown that during the electrochemical reduction of CO_2_ on the electrode, in addition to primary products, H_2_ by-products are also produced [22]. These by-products can cause side effects, such as inhibiting catalytic activity and introducing impurity doping [23]. Therefore, it is essential for the catalyst to effectively mitigate these side reactions. We calculated the specific pathways for the HER occurring on these catalysts (Figure 10). The results indicate that for both Cu_5_@GR and Cu_6_@GR, the potential barrier formed by *H is higher than the rate-limiting step for methane formation. Specifically, for Cu_5_@GR, the potential barrier is 0.348 eV versus 0.316 eV for methane formation, and for Cu_6_@GR, it is 0.389 eV versus 0.329 eV. When the external voltage is sufficient to allow the CO_2_RR to function optimally, no *H species will occupy the active sites, thus preventing interference with the CO_2_RR activity.

## 4. Conclusions

By calculating the binding energy, Cu clusters that can stably anchor on defective graphene were identified. These supported Cu clusters can serve as effective catalysts for CO_2_RR. Additionally, a projected density of states (PDOS) analysis reveals significant hybridization between the d-orbitals of Cu and the p-orbitals of C near the Fermi level, confirming the stable interaction between the copper clusters and the defective graphene surface.

A comparison of Cu_5_@GR and Cu_6_@GR as catalysts revealed that both share similar active sites and follow the same reaction pathway for CO_2_RR. The overpotentials required for CH_4_ production with these two materials are only −0.316 eV and −0.329 eV, with the rate-determining steps being *CHO → *CH_2_O and *CH_2_O → *CH_3_O, respectively.

The supported catalysts are stable and exhibit good catalytic performance, suggesting that Cu_5_@GR could be a promising candidate for CO_2_RR. It is hoped that experimental techniques can be developed to control the number and aggregation mode of Cu clusters anchored on graphene substrates, in order to meet the requirements for practical applications.

## Figures and Tables

**Figure 1 nanomaterials-15-00445-f001:**
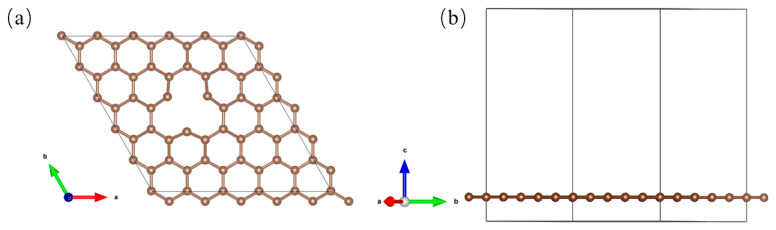
Top view (**a**) and side view (**b**) of a defective single-layer graphene.

**Figure 2 nanomaterials-15-00445-f002:**
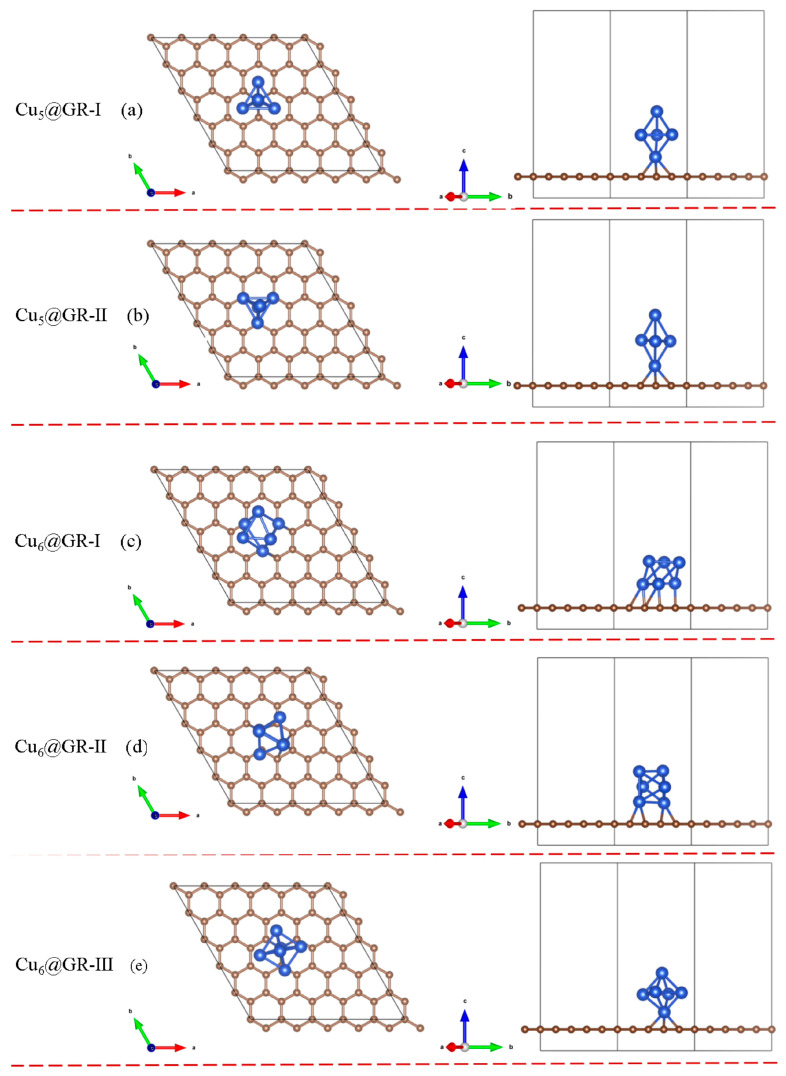
Cu*_n_*@GR Top view (**left**) and side view (**right**) of the potential configuration.

**Figure 3 nanomaterials-15-00445-f003:**
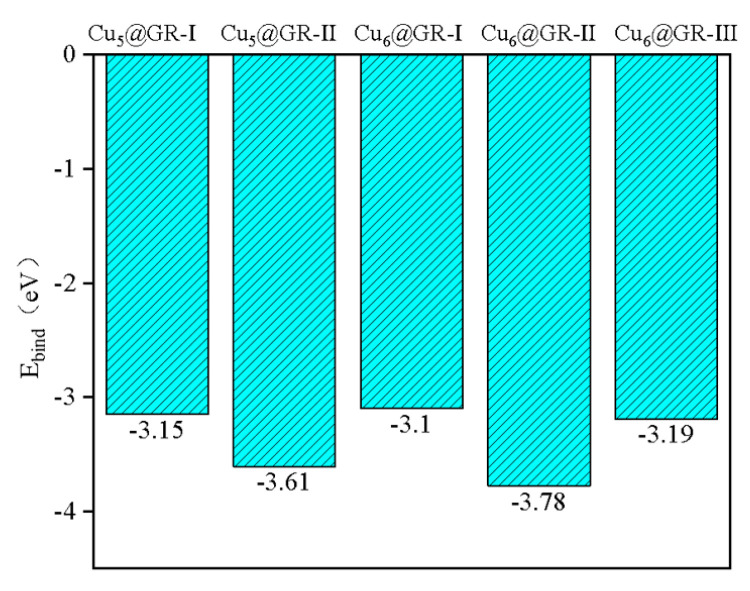
Cu*_n_*@GR Binding energy of the potential configuration.

**Figure 4 nanomaterials-15-00445-f004:**
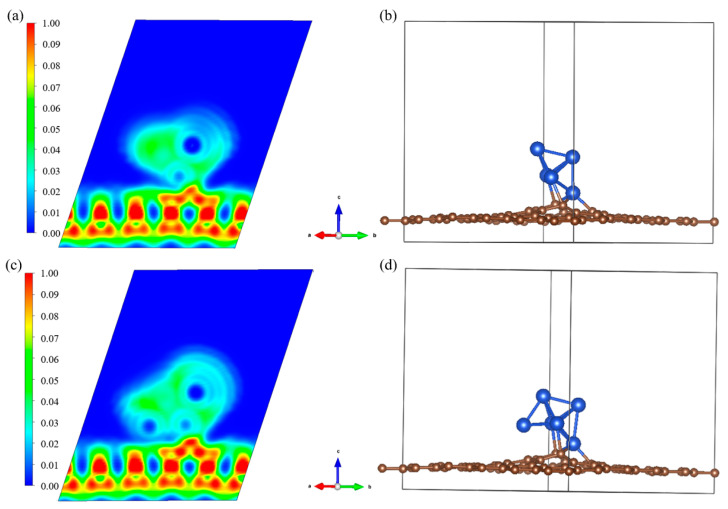
On the left are ELF maps of Cu*_n_*@GR, where the figure intercepted is the 101 cross-section of the copper cluster and the graphene substrate: (**a**) Cu_5_@GR; (**c**) Cu_6_@GR. On the right are screenshots of the structure: (**b**) Cu_5_@GR; (**d**) Cu_6_@GR.

**Figure 5 nanomaterials-15-00445-f005:**
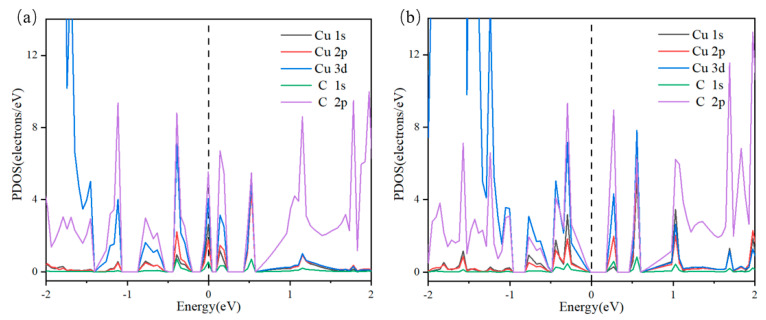
Projected density of states (PDOS). The Fermi level is referenced at 0 eV: (**a**) Cu_5_@GR; (**b**) Cu_6_@GR.

**Figure 6 nanomaterials-15-00445-f006:**
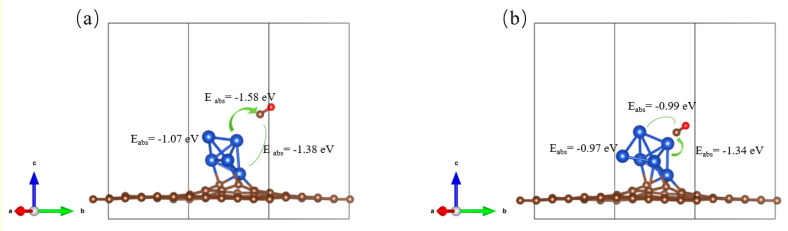
*CO Adsorption energy at various sites: (**a**) Cu_5_@GR; (**b**) Cu_6_@GR.

**Figure 7 nanomaterials-15-00445-f007:**
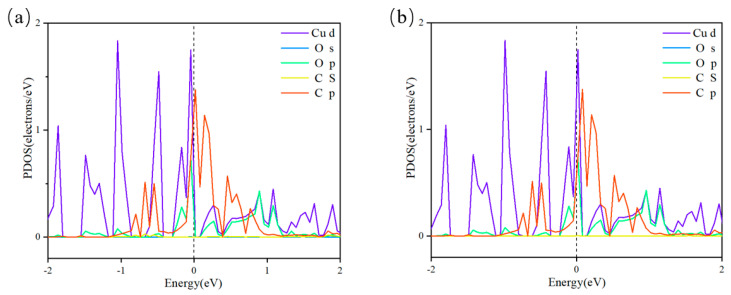
Projected density of states (PDOS) of the adsorbed *CO species on Cu*_n_*@GR. The Fermi level is referenced at 0 eV: (**a**) Cu_5_@GR; (**b**) Cu_6_@GR.

**Figure 8 nanomaterials-15-00445-f008:**
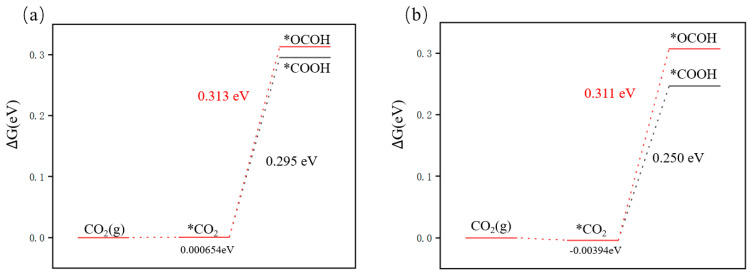
Protonation of CO_2_: (**a**) Cu_5_@GR; (**b**) Cu_6_@GR.

**Figure 9 nanomaterials-15-00445-f009:**
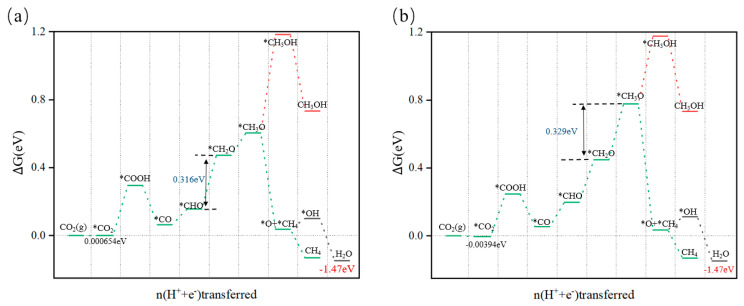
Gibbs free energy profile of the CO_2_RR pathway: (**a**) Cu_5_@GR; (**b**) Cu_6_@GR.

**Figure 10 nanomaterials-15-00445-f010:**
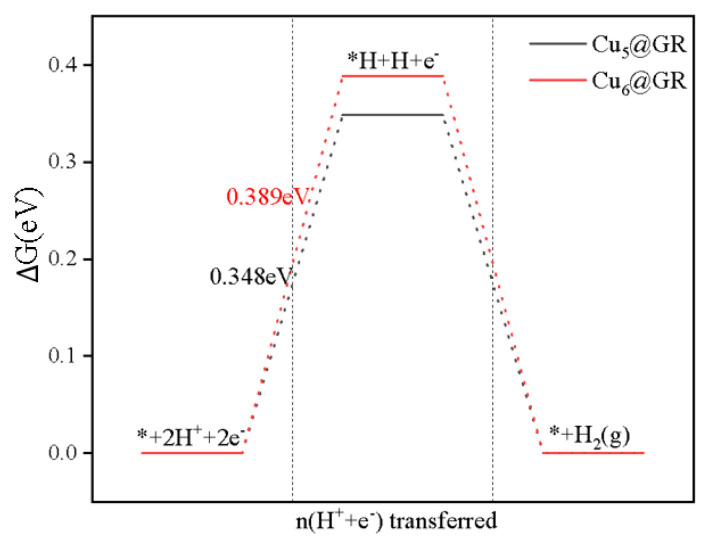
Gibbs free energy change path of HER in the CO_2_RR process at 0 V-RHE.

**Table 1 nanomaterials-15-00445-t001:** Bader charge analysis of CO_2_.

	CO_2_	CO_2_ in Cu_5_@GR	CO_2_ in Cu_6_@GR
C	1.889397	1.885506	1.895131
O-1	7.051664	7.077166	7.070804
O-2	7.058943	7.072862	7.087998

## Data Availability

The raw data required to reproduce these findings cannot be shared at this time as the data also form part of an ongoing study.

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
