# Peer review of "Graphene-Supported Cun (n = 5, 6) Clusters for CO2 Reduction Catalysis"

_nanomaterials, 2025, doi:10.3390/nano15060445_

Round 1
Reviewer 1 Report
Comments and Suggestions for Authors
Comments are attached

Reviewer 2 Report
Comments and Suggestions for Authors
Manuscript Number: Nano materials-3349501
Title: Graphene-Supported Cu (n=5,6) Clusters for CO2 Reduction 2 Catalysis
Reviewer comments
Yanling Guo et al. presents Graphene-Supported Cu (n=5,6) Clusters for CO2 Reduction 2 Catalysis. The simulation part has been discussed well. Moreover, the work was interesting and suitable for publication in the Nanomaterials. Before the article accepts for publication, the authors should address the comments listed below.
- Stability is crucial factor for electrochemical based reaction, the authors should perform stability test for of CO2
- The authors should analyze electronic structures properties of Cu (n=5,6) by using d band theory and PDOS
- Please inserts ELF map for better understanding of chemical bonding between Cu-C couple.
- The author should elaborate the underlying mechanism for selection of CO2 to CH4
- Please insert graph of charge variation or charge density difference absorbed on CO2
Reviewer 3 Report
Comments and Suggestions for Authors
This paper study the electrocatalytic CO2 reduction on Cu5/Cu6 cluster-graphene using DFT. However, there are several points that need to be clarified or supplemented:
- Regarding the free energy, the method for obtaining thermal dynamic parameters should be introduced.
- There is an obvious curvature at the graphene defect in Figure 1. Why is the graphene flat in Figure 2? Is it because the graphene model in Figure 2 is before structural optimization?
- The adsorption energy values label in Figure 4 is not clear enough, so it is difficult to draw a conclusion “Cu atoms bonded to the substrate display higher adsorption energy”. And have you considered the CO adsorption confihuration?
- Figures 5 and 6 are both directly from gas phase CO2 to hydrogenation intermediates *COOH or *OCHO, Why is there no relevant calculation of CO2 adsorption?
- what about the side reaction of HCOOH formation?
- From *CH3O to CH4, *CH3, *CH3OH, *CH4 and the desorption of CH4 and CH3OH should be added.
Round 2
Reviewer 1 Report
Comments and Suggestions for Authors
Through the review process, all necessary questions for publication have been addressed, and the manuscript is now ready to publish.
Reviewer 3 Report
Comments and Suggestions for Authors
The authors have answered all questions, this revision now is acceptable.